# Rapid and Green Preparation of Multi-Branched Gold Nanoparticles Using Surfactant-Free, Combined Ultrasound-Assisted Method

**Phat Trong Huynh** [1,2] **, Giang Dang Nguyen** [1] **, Khanh Thi Le Tran** [1] **, Thu Minh Ho** [1] **, Vinh Quang Lam** [2] **and Thanh Vo Ke Ngo** [1,*]

[1] Research Laboratories of Saigon Hi-tech Park, Ho Chi Minh City 700000, Vietnam; phat.huynhtrong@shtplabs.org (P.T.H.); giang.nguyendang@shtplabs.org (G.D.N.); khanh.tranthile@shtplabs.org (K.T.L.T.); thu.hominh@shtplabs.org (T.M.H.)

[2] Faculty of Physics and Engineering Physics, VNU.HCM, University of Science, Ho Chi Minh City 700000, Vietnam; lqvinh@vnuhcm.edu.vn

[*] Correspondence: thanh.ngovoke@shtplabs.org; Tel.: +(84)-373608890

**Abstract:** The conventional seed-mediated preparation of multi-branched gold nanoparticles uses either cetyltrimethylammonium bromide or sodium dodecyl sulfate. However, both surfactants are toxic to cells so they have to be removed before the multi-branched gold nanoparticles can be used in biomedical applications. This study describes a green and facile method for the preparation of multi-branched gold nanoparticles using hydroquinone as a reducing agent and chitosan as a stabilizer, through ultrasound irradiation to improve the multi-branched shape and stability. The influence of pH, mass concentration of chitosan, hydroquinone concentration, as well as sonication conditions such as amplitude and time of US on the growth of multi-branched gold nanoparticles, were also investigated. The spectra showed a broad band from 500 to over 1100 nm, an indication of the effects of both aggregation and contribution of multi-branches to the surface plasmon resonance signal. Transmission electron microscopy measurements of GNS under optimum conditions showed an average core diameter of $64.85 \pm 6.79$ nm and $76.11 \pm 14.23$ nm of the branches of multi-branched particles. Fourier Transfer Infrared Spectroscopy was employed to characterize the interaction between colloidal gold nanoparticles and chitosan, and the results showed the presence of the latter on the surface of the GNS. The cytotoxicity of chitosan capped GNS was tested on normal rat fibroblast NIH/3T3 and normal human fibroblast BJ-5ta using MTT assay concentrations from 50–125 µg/mL, with no adverse effect on cell viability.

**Keywords:** green synthesis; multi-branched gold nanoparticles; ultrasound; hydroquinone; chitosan





## 1. Introduction

Anisotropic gold nanoparticles are of interest in physics, chemistry, and optics, due to their unique chemical and physical properties [1–5]. In addition, they are biocompatible and less toxic [6–9]. Recently, multi-branched gold nanoparticles (GNS) attracted much attention because of their absorption in the NIR region of the electromagnetic spectrum [10]. These applications are based on the LSPR phenomenon, caused by excitation with electromagnetic radiation of the NPs [11]. The characteristics of the LSPR band, including the width, peak position, and intensity, are directly related to particles size, morphology, and properties of capping agents [12]. Depending on the nature of the GNS, the absorption spectrum is composed of a weak band at 520–550 nm and a broad band between 700–1100 nm [10]. The band at the shorter wavelengths is attributed to the transverse plasmon resonance, whereas the longitudinal component is at the longer wavelengths. Due to the shape of the anisotropic particles, there are contributions to the plasmon resonance spectrum from both the transverse and longitudinal directions, with the latter being

predominant, as the core size decreased and the length of the arms increased. However, in circumstances where the core diameters of the particles are longer than the arms, the band broadening could be due to particle aggregation, where the neighbor core plasmon resonance interact. Highly branched gold nanoparticles with small cores are characterized by broad and red-shifted LSPR maximum, with very weak or absent bands between 520–550 nm.

Multi-branched gold nanoparticles or gold nanostars are usually synthesized by the seed-mediated method, in the presence of anionic or cationic surfactants [13,14]. They take the role of shape-directing and capping agents to drive structural growth and prevent the aggregation of gold nanoparticles. The presence of surfactants result in cellular toxicity and they are also difficult to remove before the particles are used [15,16]. Many reports of surfactants-free GNS in recent years [17,18] are mainly based on seed-mediated protocols. The use of sodium borohydride, hydrazine, or trisodium citrate in seed-mediated synthesis of GNS, results in cytotoxicity and they are harmful to the environment.

Recently, green chemistry has become a popular trend in a variety of fields, as it offers a number of advantages, including safety, energy efficiency, and the production of less toxic waste [19–21]. In green synthesis of nanoparticles, people evaluate and select new nontoxic reductants, protecting agents, innocuous solvents from nature (leaf extract, microorganism, or nature polymer) to replace toxic materials [22,23], and develop advanced and energy-efficient techniques such as microwave and sonochemical methods [24,25].

Sonochemical effects are caused by acoustic cavitation in liquids with the creation and collapse of bubbles. The collapse of the cavitation bubbles is more rapid than thermal transport and generates "localized hot spots", which have a temperature of 5000 K, pressures of about 2000 atm and cooling rates of more than 109 K/s [26]. The advantages of using the sonochemical approach include the production of high purity, uniform shape, high yields, and cost effective synthesis of nanoparticles.

Several one-pot synthesis techniques of multi-branched gold nanoparticles were studied [27–29]. The advantages of these procedures are facile and free-surfactant synthesis but just short-branches gold nanostars were formed (less than 10 nm branches). Sonochemical synthesis is a popular method for spherical nanoparticles [30]. However, it is rarely studied using ultrasound assist to synthesize gold nanostars, except reports of Badilescu et al. [31]. In this work, we introduce a rapid and green preparation of multi-branched gold nanoparticles using surfactant-free and seedless combined ultrasound (US) assisted protocol. Chitosan (CS) was used as a stabilizer while hydroquinone (HQ) was used as the reducing agent. CS is a polysaccharide derived from shrimps, crabs, and other crustaceans. Due to its many advantageous properties such as biocompatibility, nontoxicity, low-cost, biodegradability, and antimicrobial agent, CS has a number of commercial and biomedical uses. Additionally, HQ has a variety of uses as a reducing agent that is soluble in water. Furthermore, HA was used to replace a traditional reducing agent, ascorbic acid, which could tune the adsorption of anisotropic gold nanoparticles towards the far NIR region [32]. Additionally, sonochemical or ultrasound assistance was applied to control the size and shape, and enhance the stability of GNS.

## 2. Materials and Methods

### 2.1. Materials

Chloroauric acid (HAuCl$_4$xH$_2$O, ~52% Au basis), sodium hydroxide 97%; chitosan (low molecular weight), hydroquinone 99%, phosphate buffer saline (tablet, pH 7.2–7.6), acid acetic 99% were purchased from Sigma-Aldrich, St. Louis, Missouri, US. The materials for cytotoxicity include Dulbecco's Modified Eagle Medium (DMEM), Fetal Bovine Serum (FBS), Bovine Calf Serum (BCS) were supplied by Sigma-Aldrich, Darmstadt Germany. Thiazolyl blue tetrazolium bromide (MTT), and antibiotic solution (penicillin-streptomycin) were supplied by Sigma-Aldrich, St. Louis, Missouri, US. Lysis buffer solution was purchased from Biobasic, Markham, Ontario, Canada. Normal mouse fibroblast (NIH/3T3,

CRL-1658) and normal human fibroblast (BJ-5ta, CRL-4001) were ordered from ATCC, Manassas, Virgina, US. Deionized (DI) water 17.8 MΩ was used throughout the experiments.

*2.2. Methods*

2.2.1. Surfactant-Free Preparation of Multi-Branched Gold Nanoparticles (GNS) by the One-Step Method

First, 2 g chitosan (CS) powder was homogenized into a 98 g acid acetic 1% solution to make CS 2% solution. Next, 1 mL of $HAuCl_4$ $2 \times 10^{-2}$ M solution was added to 9 mL chitosan solution, under stirring. This mixture was pH-adjusted by acetic acid. Finally, hydroquinone (HQ) 0.1 M was mixed immediately into this mixture. The reaction solution was kept constant for 30 min at room temperature. Throughout the period of reaction, the color of the solution changed from colorless to cobalt blue, indicating that multi-branched particles had formed. The influences of conditions including pH, mass concentration of CS, hydroquinone concentration, and morphology of GNS were studied.

2.2.2. Surfactant-Free Preparation of GNS Combined Ultrasound

The procedure of preparation for the combined ultrasound (US) was similar to the process above, except that the reaction solution was sonicated instead of being kept at room temperature without stirring. Sonication was carried out using the Q2000 sonicator—Qsonica, Newtown, Connecticut, US (power 1.375 watts, frequency 20 kHz). Chitosan-coated GNS was synthesized at a constant frequency of 20 kHz and six different levels of amplitude (0, 20, 40, 60, 80, and 100 µm) and six time-periods (0, 2, 4, 6, 8, and 10 min), in order to investigate the effect of amplitude and sonication time on size, shape, and stability of GNS.

2.2.3. Characterization

UV–Vis spectrophotometer Dynamica Halo RB-10 (Dynamica, Livingston, UK) was used to record the surface plasmon resonance (SPR) of GNS in the wavelength range of 400–1100 nm, at a scanning rate of 200 nm/min. The interaction between GNS and CS was shown by FT-IR analysis (Bruker Tensor 27, Bruker Optics, Ettlingen, Germany). All FT-IR results were obtained from the powder samples and smoothing or correction baseline was not applied. The crystal structure of GNS was determined by employing X-ray diffraction (XRD). Scanning was carried out in the 2 theta range of 20–100°, using the X-ray diffractometer Bruker D5005 (Bruker AXS, Karlsruhe, Germany). Transmission electron microscope (TEM) analysis was examined by JEM1010-JEOL (Jeol, Tokyo, Japan). The J-Image software (NIH Image) was used to calculate the average length of branches as well as diameter of cores of GNS, based on thirty particles of each three sample from the TEM images. All analyses including UV-Vis, FT-IR, XRD, and TEM, as well as the cytotoxicity test below were carried out through separation from the same sample. The GNS solutions were sonicated before examinations and measurements.

2.2.4. Cytotoxicity of Chitosan-Capped GNS

NIH/3T3 were cultured in DMEM (Dulbecco's Modified Eagle Medium) supplemented with 1% penicillin–streptomycin (Pen–Strep) and 10% bovine calf serum (BCS), while the BJ-5ta cells were in DMEM with 1% Pen–Strep and 10% FBS (Fetal Bovine Serum. Both cell lines were grown in a humidified incubator with 5% $CO_2$. The cell lines were detached from the culture flasks using a trypsin (0.25%)—EDTA (0.53 mM) solution. Cell viability was evaluated using thiazolyl blue tetrazolium bromide (MTT). Both cell lines were seeded onto 96 well plates at $10^4$ cells/well. CS-coated GNS was added to each well so that the final concentration samples were 50, 75, 100 and 125 µg/mL. The negative control subject was Lysis buffer. Cells were incubated for 24 h in an incubator (37 °C and 5% $CO_2$). Then, MTT solution (0.5 mg/mL) was added to each well. The plates were incubated at 37 °C for 4 h to form MTT formazan. Lysis buffer (4 mM HCl) was added to each well to dissolve the crystallized MTT formazan.

The plate absorbance (OD) was read at wavelength λ = 570 nm, using a Microreader. Cell viability could be calculated and compared to the control samples, as follows:

$$\% \text{ cell viability} = \frac{OD \ of \ sample}{OD \ of \ negative \ control} \times 100\% \tag{1}$$

## 3. Results and Discussion

### 3.1. One-Step Surfactant-Free Preparation of GNS

3.1.1. Effect of Mass Concentration of CS

Figure 1 shows that the UV–Vis result of GNS prepared in different mass concentration of CS. According to previous reports, the absorption spectrum was a plasmon resonance peak (SPR) ranging from 500 to over 1100 nm, indicating multi-branched particle presence. The SPR absorbance of GNS shifted, depending on the size and shape [33]. An increase in length of branches led to SPR shift toward the NIR region [34], while SPR absorbance moved to blue shift, due to the short branched formations [35]. On the other hand, broadening and increasing the maximum absorbance wavelength of SPR toward the NIR region suggested a form of aggregation, due to the interaction of the spherical core of the particles with each other [36]. It was clear that the intensity of SPR increased when the mass concentration of CS was increased from the beginning, evaluating the concentration to 1%. However, the SPR absorbance moved to the blue shift and the intensity of SPR decreased when mass concentration increased to 2%. This could be related to an increase in the size of the core particles. Table 1 describes the influence of % CS to SPR and intensity of SPR of GNS. At the beginning, 0.25% CS both of the SPR and the intensity of SPR rose from 845 nm and 0.15, to 875 nm and 0.18, respectively. With the increasing % CS, although the SPR increased slightly to 878, the intensity of SPR reached the highest value at 0.33. However, both SPR dropped to 667 nm and 877 nm, when % CS increased to 2. TEM images of GNS are shown in Figure 2. The prepared GNS at 1% CS (Figure 2a) had an average core of 57.33 ± 5.91 nm, and long, sharp branches with an average length of 44.32 ± 9.27 nm. Meanwhile at 1.5% CS, the average length of the branches decreased to 20.71 ± 8.57 nm (Table 2).

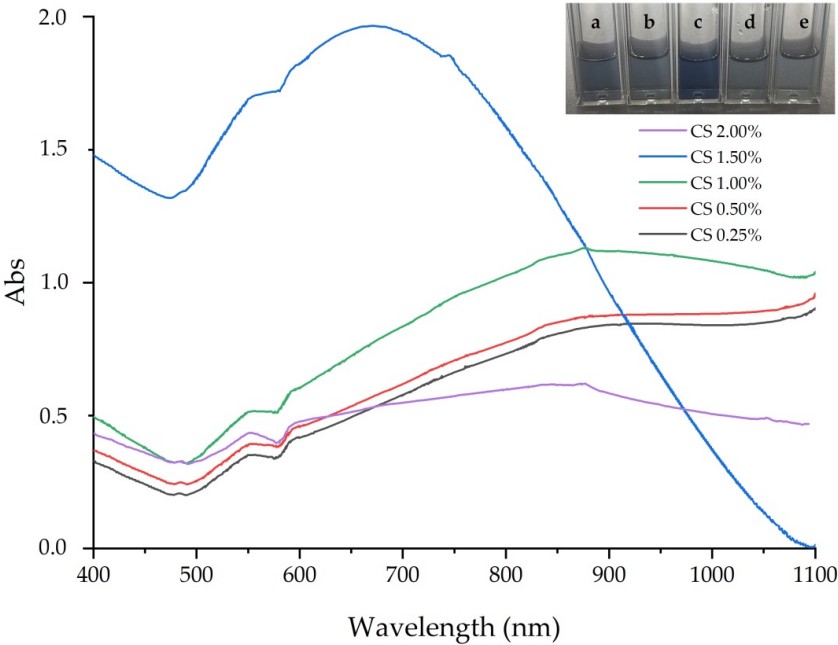

**Figure 1.** Absorption spectra of GNS prepared in various mass concentrations of CS: (**a**) 0.25%, (**b**) 0.50%, (**c**) 1.0%, (**d**) 1.5% and (**e**) 2.0%.

**Table 1.** The influence of mass concentration of chitosan to surface plasmon resonance and intensity absorbance.

| % CS (w/v) | SPR (nm) | Int. of SPR |
|---|---|---|
| 0.25 | 845 | 0.15 |
| 0.50 | 875 | 0.18 |
| 1.00 | 880 | 0.33 |
| 1.50 | 667 | 1.04 |
| 2.00 | 877 | 0.21 |

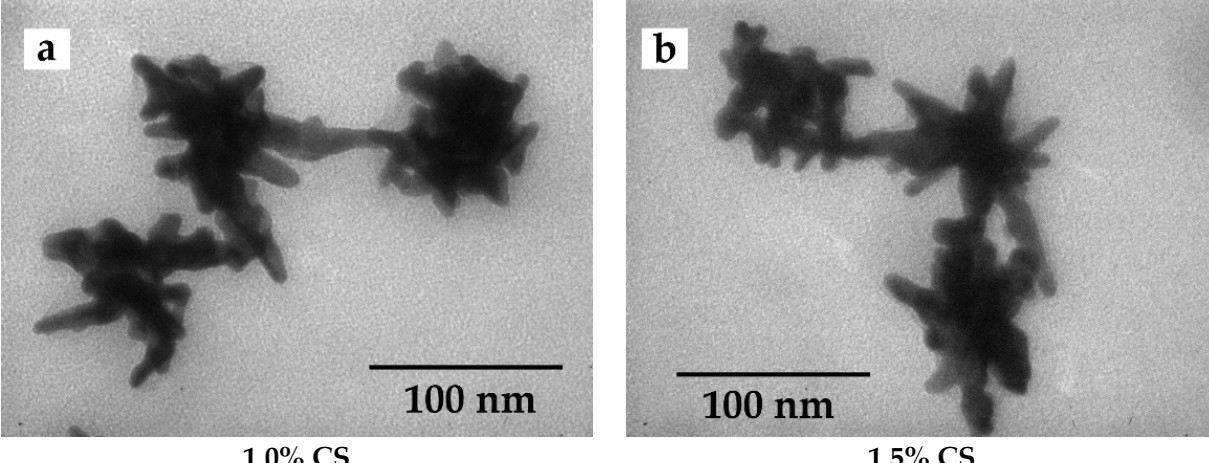

**1.0% CS**                    **1.5% CS**

**Figure 2.** TEM images of GNS prepared in 1.0% (**a**) and 1.5% CS (**b**).

**Table 2.** The influence of mass concentration of chitosan on the morphology, average core and branches of GNS.

| % CS (w/v) | Morphology | Core (nm) | Branches (nm) |
|---|---|---|---|
| 1.00 | Long multi-branches | 57.33 ± 5.91 | 44.32 ± 9.27 |
| 1.50 | Short multi-branches | 51.38 ± 9.14 | 20.71 ± 8.57 |

### 3.1.2. Effect of pH

The role of pH on the formation of GNS was illustrated by UV–Vis absorption spectra (Figure 3) and TEM images (Figure 4). There was a rise of SPR absorbance and intensity of SPR, as pH 1.0 was adjusted to pH 1.5 and reached the highest value, 865 nm of absorbance and 0.49 of intensity. The SPR and intensity of SPR significantly declined when pH was adjusted to 2.0, 2.5, and finally 3.0. The SPR dropped to the lowest peak 833 nm, while intensity went down to 0.04, following pH adjustment (Table 3). As per the UV–Vis spectra and TEM examinations, the effect of both size of core and branches of multi-branched particles on SPR absorbance was shown. Adjusting pH from 1.0 to 2.0, the SPR absorbance spectra shifted to the NIR region and SPR intensity declined from 0.54 to 0.16. These were contributed by an increase of size of core and prolongated branches (Table 4). At pH 1.0, the GNS formed were short, multi-branch particles with 53.44 ± 8.30 nm average core and 36.23 ± 8.84 nm branches (Figure 4a). At pH adjusted to 1.5, the prepared GNS particles had elongated branches of 45.23 ± 10.03 nm (Figure 4a) and a core diameter of 59.32 ± 8.08 nm. Meanwhile, when the pH increased to 2.0, and the branches of GNS were short and unsharp (Figure 4b). At pH 3.0, a just, non-star shape were formed (few branches particles) and aggregation of the core of nanoparticles was observed from Figure 4c. Both TEM images and decrease in SPR intensity indicated that there was a greater contribution to the adsorption spectrum because of the core aggregation than the surface plasmon resonance from multibranches, at basic pH. The influence of pH to

formation mechanism of gold nanoparticles could be explained by the relation between pH and HQ, as a reducing agent [37]. HQ can reduce $Au^{3+}$ ions to $Au^0$ atoms by electrons produced in the oxidation/reduction process. Oxidation/reduction is a reversible process at equilibrium. At high acidic conditions (pH below 1.0), HQ could not reduce $Au^{3+}$ to Au atoms, in the absence of gold seeds, because there were not any produced electrons. Adjusted to pH 1.5, few gold seeds appeared due to rapid reduced $Au^{3+}$ ions. Furthermore, the reversible reaction promoted some electrons that reduced $Au^{3+}$ to $Au^+$ ions. Under protection of CS, $Au^+$ attached to the gold seeds, and grew in anisotropic directions to form multi-branched particles. However, at pH towards basic condition (pH upper 2.0), the process became irreversible and was driven to the right. This promoted numerous electrons and uncontrolled gold nanoparticles were synthesized.

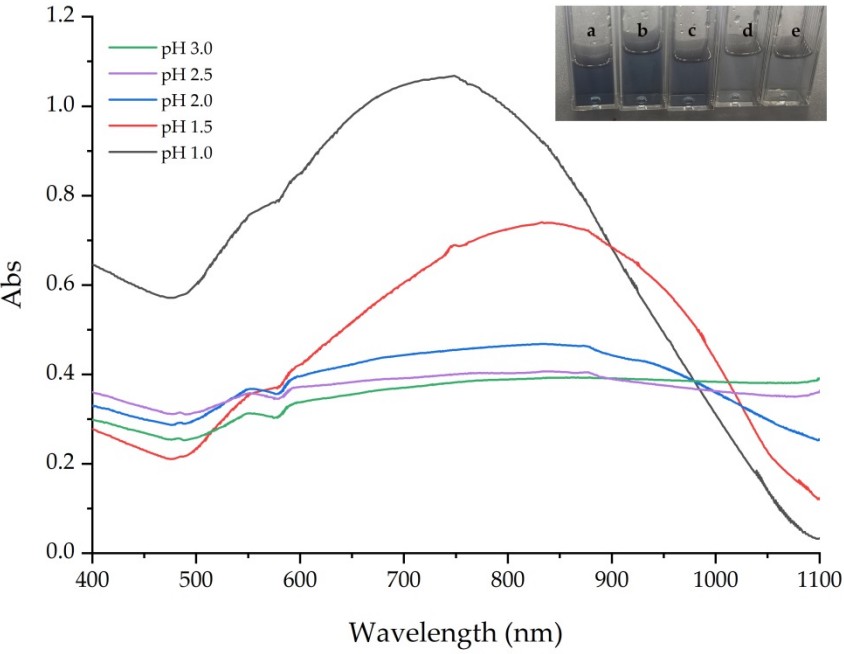

**Figure 3.** Absorption spectra of GNS prepared in different pH: (**a**) 1.0, (**b**) 1.5, (**c**) 2.0, (**d**) 2.5 and (**e**) 3.0.

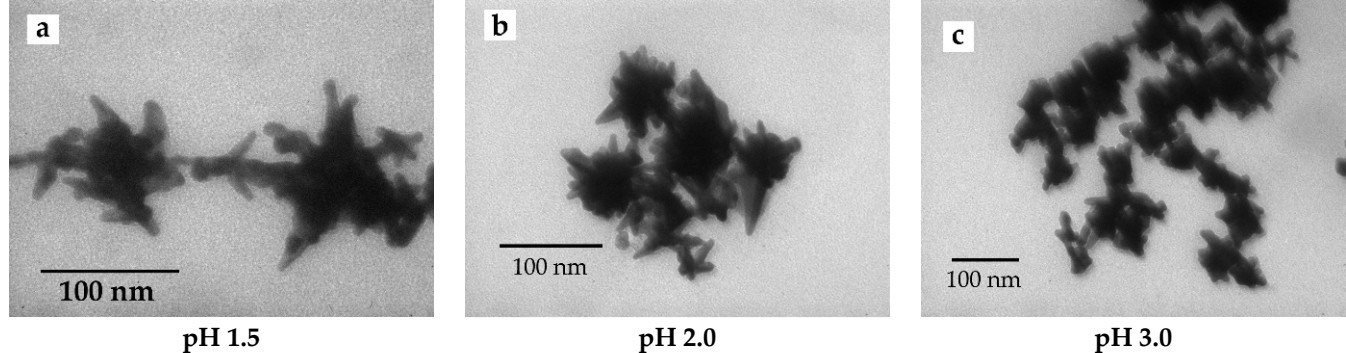

**Figure 4.** TEM images of GNS prepared in different pH: (**a**) 1.5, (**b**) 2.0, and (**c**) 3.0.

**Table 3.** The influence of pH to surface plasmon resonance and intensity absorbance.

| pH | SPR (nm) | Int. of SPR |
|----|----------|-------------|
| 3.0 | 833 | 0.04 |
| 2.5 | 834 | 0.06 |
| 2.0 | 835 | 0.16 |
| 1.5 | 865 | 0.49 |
| 1.0 | 746 | 0.54 |

**Table 4.** The influence of pH to morphology, average core and branches of GNS.

| pH | Morphology | Core (nm) | Branches (nm) |
|----|------------|-----------|---------------|
| 1.0 | Long multi-branches | 53.44 ± 8.30 | 36.23 ± 8.84 |
| 1.5 | Long multi-branches | 59.32 ± 8.08 | 45.23 ± 10.03 |
| 2.0 | Short multi-branches | 55.66 ± 12.28 | 19.55 ± 9.60 |
| 3.0 | Few branches | 99.89 ± 16.84 | 34.83 ±13.22 |

### 3.1.3. Effect of Hydroquinone

Different volumes of HQ 0.1 M were adjusted to investigate the effect of HQ for preparation of GNS. The absorption spectra of GNS prepared in a variety of HQ volume are displayed in Figure 5. There was an increase of intensity of SPR, from 0.40 to 1.16, and the SPR absorbance also shifted to 708 nm, as 1.0 to 2.0 mL of HQ volume was added (Table 5). The intensity of SPR declined rapidly to 0.67 and 611 nm of SPR absorbance, when amounts of 2.0 mL to 3.0 mL HQ 0.1 M were added, respectively. TEM micrographs in Figure 6 revealed morphology of three samples of GNS. At low volume of HQ (2.0 mL or less), GNS exhibited short multi-branched particles of 19.58 ± 5.19 nm, with an average core of 51.22 ± 6.67 nm. The 2.0 mL HQ volume resulted in long, sharp branches of GNS. The average branched length was 76.11 ± 14.23 nm and the core was 64.85 ± 6.79 nm. However, the branches of GNS were shorter when the HQ volume increased to 3.0 mL. This resulted in 55.15 ± 10.68 nm average core and 49.85 ± 7.42 nm in branched length (Table 6). Additionally, TEM revealed to the interaction of core particles. The influence of volume of HQ 0.1 M could be explained as follows. Gold seeds were first formed, then $Au^{3+}$ ions were reduced to $Au^+$. Then, the $Au^+$ ions attached to the gold seeds and grew in anisotropic directions to form multi-branched particles. At low HQ, the electrons produced by the reversibility of HQ was enough to promote seed-particles, meanwhile small amounts of $Au^+$ ions appeared. Therefore, the short multi -branched particles were synthesized. At high HQ, not only seed particles formed but also all $Au^{3+}$ changed to $Au^+$ ions, resulting in the formation of long and sharp multi-branched particles. However, if the HQ was too high, many electrons were promoted, which led to an uncontrolled reaction [38].

**Table 5.** The influence of HQ to surface plasmon resonance and intensity absorbance.

| Volume HQ 0.1 M (mL) | SPR (nm) | Absorbance |
|----------------------|----------|------------|
| 1.0 | 630 | 0.40 |
| 1.5 | 746 | 0.64 |
| 2.0 | 708 | 1.16 |
| 2.5 | 652 | 0.77 |
| 3.0 | 611 | 0.67 |

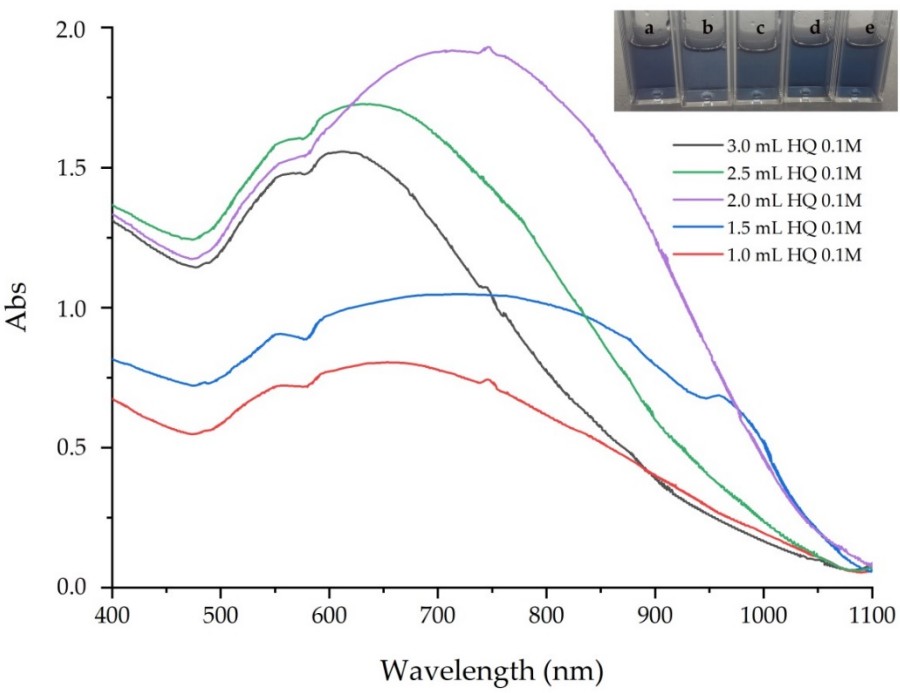

**Figure 5.** Absorption spectra of GNS prepared in various HQ 0.1 M volume—(**a**) 1.0 mL, (**b**) 1.5 mL, (**c**) 2.0 mL, (**d**) 2.5 mL and (**e**) 3.0 mL.

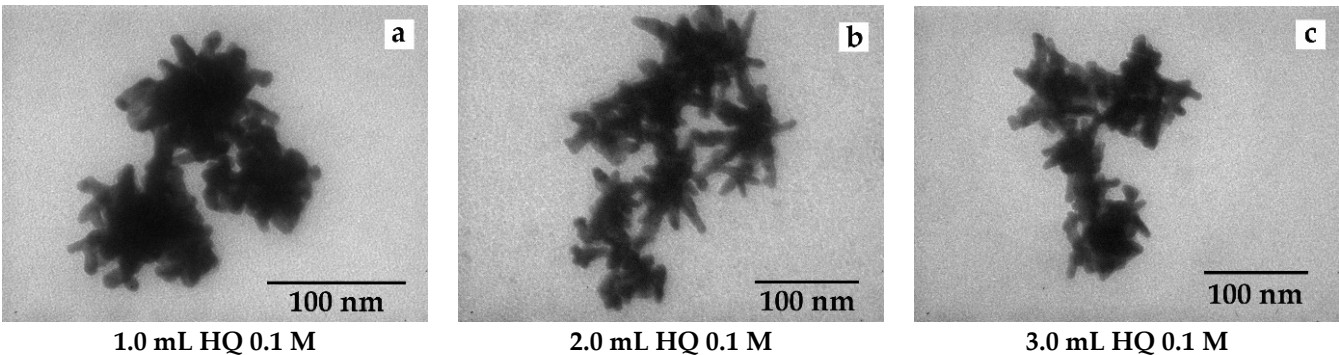

**Figure 6.** TEM images of GNS prepared in various HQ 0.1 M volume—(**a**) 1.0 mL, (**b**) 2.0 mL, and (**c**) 3.0 mL.

**Table 6.** The influence of HQ on morphology, average core and branches of GNS.

| Volume HQ 0.1 M (mL) | Morphology | Core (nm) | Branches (nm) |
|:---:|:---:|:---:|:---:|
| 1.0 | Short multi-branches | $51.22 \pm 6.67$ | $19.58 \pm 5.19$ |
| 2.0 | Long multi-branches | $64.85 \pm 6.79$ | $76.11 \pm 14.23$ |
| 3.0 | Long multi-branches | $49.85 \pm 7.42$ | $55.15 \pm 10.68$ |

### 3.2. Ultrasound Combined Surfactant-Free Preparation of GNS

#### 3.2.1. Effect of Sonication Amplitude

UV–Vis results (Figure 7) and TEM analysis (Figure 8) were used to study the effect of amplitude sonication on the morphology and size of GNS synthesized in permanent time sonication, for 5 min. It was noticeable that the intensity of SPR of sonicated GNS was more intent than without the sonication handled sample. The intensity of SPR increased from 0.26 to 0.53, while the SPR absorbance rose from 831 nm to 877 nm, due to the changing amplitude from 20 to 60 μm (Table 7). Adjusting the amplitude to 80 μm, the branches of multi-branched particles shortened dramatically to $27.88 \pm 5.83$ nm, while the core diameter increased to $73.10 \pm 24.66$ nm. In addition, the aggregation of multi-

branched particles was revealed due to the interaction of the spherical core with each other. Additionally, both SPR absorbance and intensity of SPR still had a high value, 874 nm and 1.14, respectively. This could be assigned to the greater contribution of the UV–Vis spectrum because of aggregation rather than the formation of surface plasmon resonance branches. The intensity of SPR significantly declined to 0.80, when amplitude was adjusted to d 100 μm, and the SPR absorbance fell to 690 nm. Synthesized GNS in 60 μm amplitude sonication have long and sharp multi-branches, whereas in prepared GNS in higher amplitude, the branches were short and un-harp (Table 8). Sonication power is the electrical energy supplied to the probe sonicator and transformed into mechanical energy. This is executed by exciting the piezoelectric crystals moving in the longitudinal direction, where the mechanical energy result in the probe vibrating up and down [39]. The stronger amplitude applied, the higher is the acoustic energy produced. When suitable amplitude sonication was applied, the reduction of $Au^+$ to $Au^0$ were accelerated, which resulted in higher reaction yields than without sonication-assisted sample, in the same time reaction. Nevertheless, an overly strong amplitude generated a high acoustic energy that led to uncontrolled rapid reduction, so the overall isotropic small particles reaction dominated the over multi-branched anisotropic.

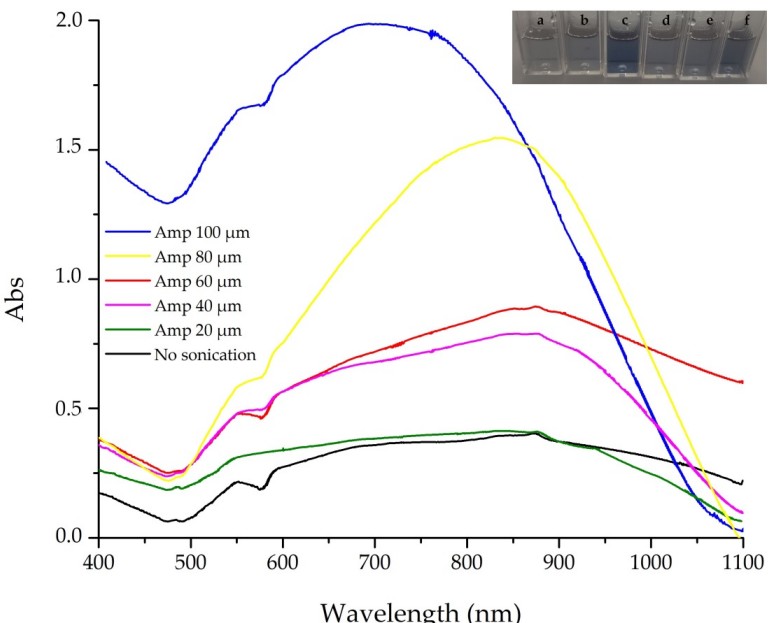

**Figure 7.** Absorption spectra of GNS prepared in different amplitudes—(**a**) 0 μm, (**b**) 20 μm, (**c**) 40 μm, (**d**) 60 μm, (**e**) 80 μm and (**f**) 100 μm.

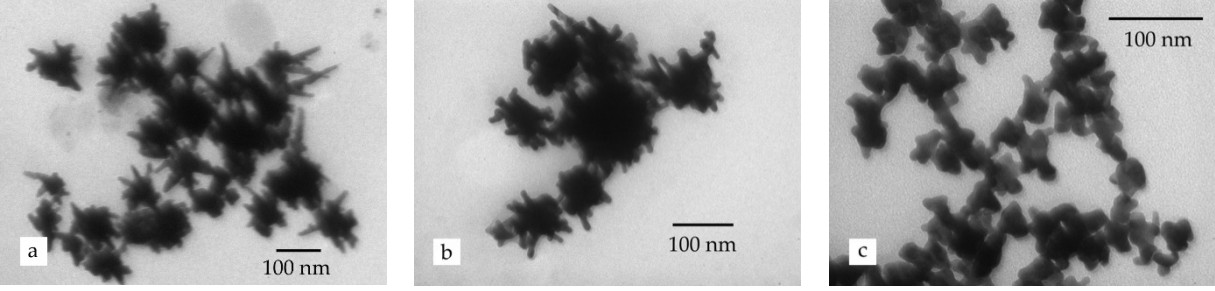

**Figure 8.** TEM images of GNS prepared in different amplitudes—(**a**) 60 μm, (**b**) 80 μm and (**c**) 100 μm.

**Table 7.** The influence of amplitude to surface plasmon resonance and intensity absorbance.

| Amplitude (µm) | SPR (nm) | Absorbance |
|:---:|:---:|:---:|
| 0 | 831 | 0.20 |
| 20 | 832 | 0.26 |
| 40 | 874 | 0.53 |
| 60 | 877 | 0.37 |
| 80 | 874 | 1.24 |
| 100 | 690 | 0.80 |

**Table 8.** The influence of amplitude sonication to morphology, average core and branches of GNS.

| Amplitude (µm) | Morphology | Core (nm) | Branches (nm) |
|:---:|:---:|:---:|:---:|
| 60 | Long multi-branches | $62.08 \pm 7.32$ | $67.73 \pm 22.73$ |
| 80 | Short multi-branches | $73.10 \pm 24.66$ | $27.88 \pm 5.83$ |
| 100 | Few branches | $28.26 \pm 3.66$ | $8.99 \pm 2.58$ |

3.2.2. Effect of Sonication Time

Figure 9 (the UV–Vis results) and Figure 10 (TEM images) show the influence of time sonication on the morphology and size of GNS prepared with a constant amplitude 50 µm. Both intensity and SPR absorbance increased when a longer sonication time was applied, and it reached the highest value at 6 min. Nevertheless, both decreased rapidly, despite a prolonged time sonication. Table 9 indicates that the intensity from 0.32 rose to 1.15 and SPR absorbance from 831 nm shifted to 833 nm, with an elongated time sonication of 6 min. The SPR absorbance shifted to the NIR region because both the size of core and the branched length increased. Synthesized GNS at 4 min exhibited an average branched length of $31.32 \pm 7.62$ nm and a core diameter of $52.02 \pm 9.95$ nm, that from 6 min had branches of $65.01 \pm 11.39$ nm and an average core of $75.34 \pm 18.37$. Intensity of SPR dropped to 0.44, while the SPR shifted significantly down to 630 nm. The time sonication of 8 min resulted in GNS with short and unsharp branches, besides, there was a rise of core diameter of multi-branched particles due to the interaction of each particle. Meanwhile, longer time sonication obtained gold nanoparticles that were short rod particles (Table 10). It is possible that the influence of time sonication might be based on cavitation bubbles. Sonication for a prolonged period that generated more cavitation bubbles led to an uncontrolled and rapid reduction, due to higher acoustic energy. As a result, the gold nanoparticles had a smaller core, and fewer and shorter branches were formed [40].

**Table 9.** The influence of time sonication to surface plasmon resonance and intensity absorbance.

| US Time (min) | SPR (nm) | Absorbance |
|:---:|:---:|:---:|
| 0 | 831 | 0.32 |
| 2 | 833 | 0.39 |
| 4 | 832 | 1.13 |
| 6 | 833 | 1.15 |
| 8 | 746 | 0.96 |
| 10 | 630 | 0.44 |

*3.3. Investigation of Interaction between CS and GNS*

The FT–IR of pure CS and CS-coated GNS spectra are shown in Figure 11. In the spectrum of pure CS, there was a band at around 2883 cm$^{-1}$ that corresponded to the stretching vibration of the C-H groups. Two peaks located at 1649 cm$^{-1}$ and 1324 cm$^{-1}$ related to the C=O stretching vibration of amide I and C-N stretching of amide III, respectively [41]. These bands confirmed the N-acetyl groups of CS [42]. The band at 1589 cm$^{-1}$ corresponded to the bending vibration of the N-H groups (amide II) [42].

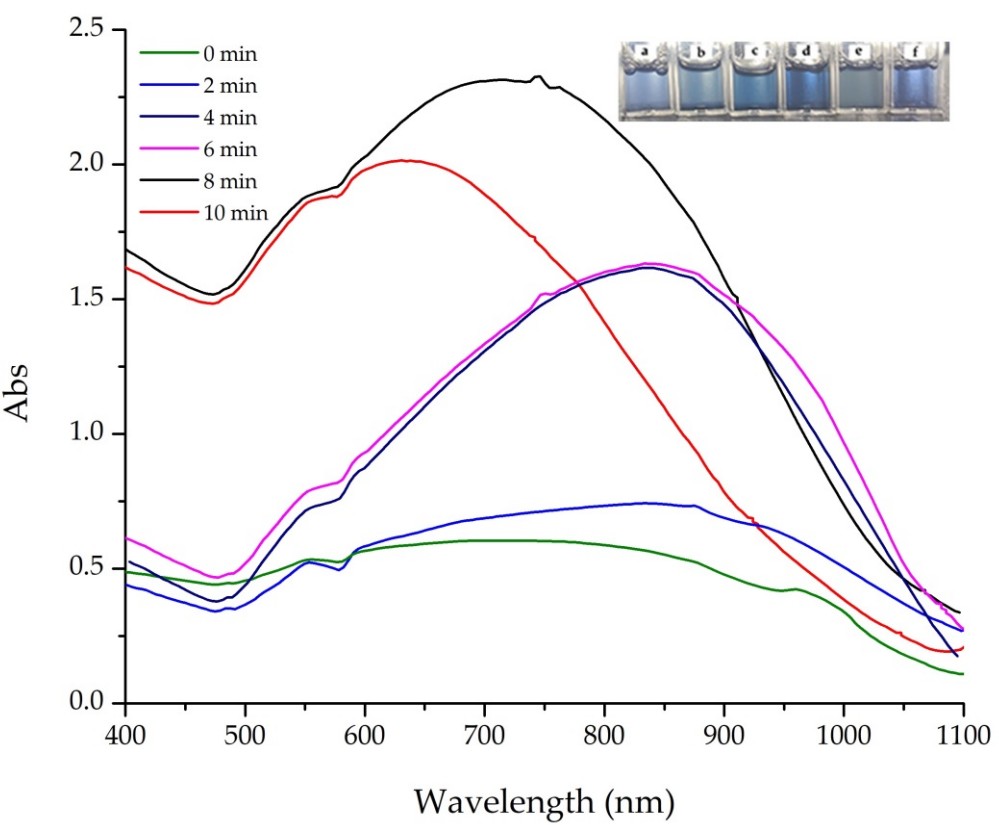

**Figure 9.** Absorption spectra of GNS prepared in different sonication time—(**a**) 0 min, (**b**) 2 min, (**c**) 4 min, (**d**) 6 min, (**e**) 8 min and (**f**) 10 min.

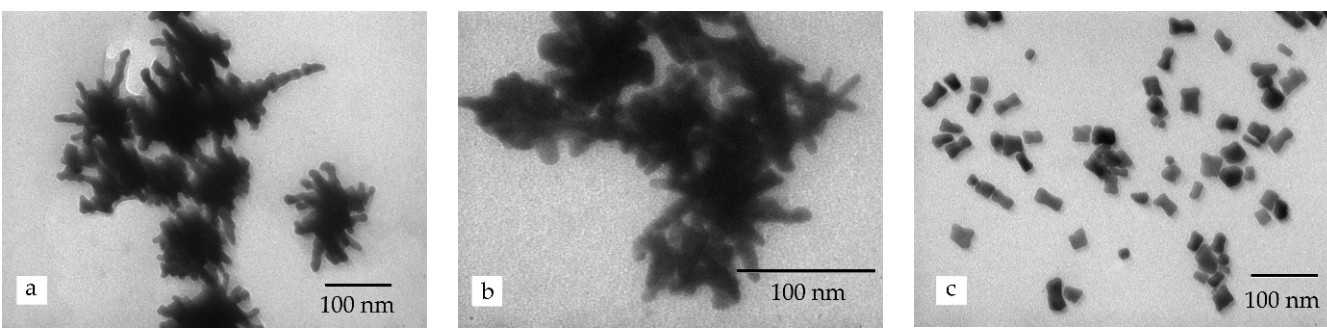

**Figure 10.** TEM images of GNS prepared in different sonication time—(**a**) 6 min, (**b**) 8 min and (**c**) 10 min.

**Table 10.** The influence time sonication to morphology, average core and branches of GNS.

| US Time (min) | Morphology | Core (nm) | Branches (nm) |
| :---: | :---: | :---: | :---: |
| 4 | Long multi-branches | $52.02 \pm 9.95$ | $31.32 \pm 7.62$ |
| 6 | Long multi-branches | $75.34 \pm 18.37$ | $65.01 \pm 11.39$ |
| 8 | Short multi-branches | $87.44 \pm 11.08$ | $18.65 \pm 5.01$ |
| 10 | Short rod-shape | - | - |

The FT–IR spectrum of CS-capped GNS was the shift of bands observed in pure CS. The C-H stretching shifted to 2880 cm$^{-1}$, while the bending of N-H bonds moved to 1557 cm$^{-1}$. Additionally, stretching of C=O (amide I) and C-N (amide III) was located at 1642 cm$^{-1}$ and 1310 cm$^{-1}$. These shifts indicated the interaction between the functional groups of CS and GNS [42].

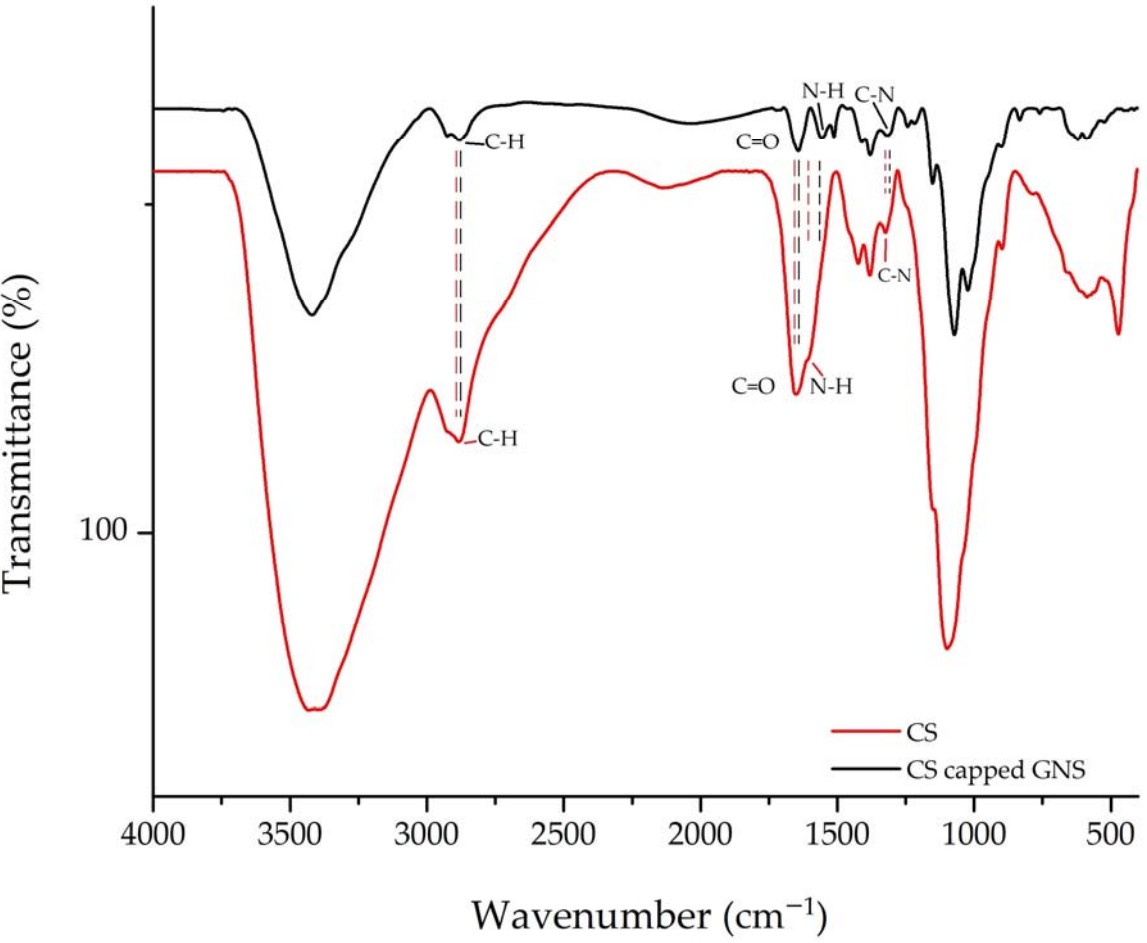

**Figure 11.** FT–IR spectrum of CS and CS-coated GNS.

### 3.4. The XRD Diagram of GNS

Figure 12 is the XRD diagram of CS-coated GNS. The recorded pattern exhibited peaks located at 38.0°, 44.9°, 65.2° and 77.4°, which correspond to the (111), (200), (220), (311) and (222) planes of the gold face-centered-cubic (fcc) crystalline structure, respectively [43]. It was clear that there was an intense peak located at 38.0°, which was indexed to the (111) plane. Additionally, a weaker peak for the (200) plane at 65.2° and another at 77.3° for the (311) plane was observed. Finally, a very weak peak located at 77.3° related to the (311) plane.

### 3.5. Cytotoxicity of the CS-Capped GNS

Biocompatibility is an important property for biomedical applications. MTT assay was used to study the cell compatibility of CS-coated GNS. The cell viability of normal rat fibroblast (NIH/3T3) and normal human fibroblast (BJ-5ta), exposed to various concentrations of multi-branched nanoparticles are shown in Figure 13. The proliferation of both NIH/3T3 and BJ-5ta cell lines treated in various GNS concentration were still around 90%, even at a high concentration of 200 µg/mL, except for BJ-5ta, which had a concentration of over 82%. The results indicate that CS-coated GNS was a good biocompatible agent and is a prospective material for use in biomedical applications. GNS have wide biomedical applications in Raman scattering sensing [44], stem cell tracking [45], bioimaging [46], photothermal treatment [47] and immunotherapy [48].

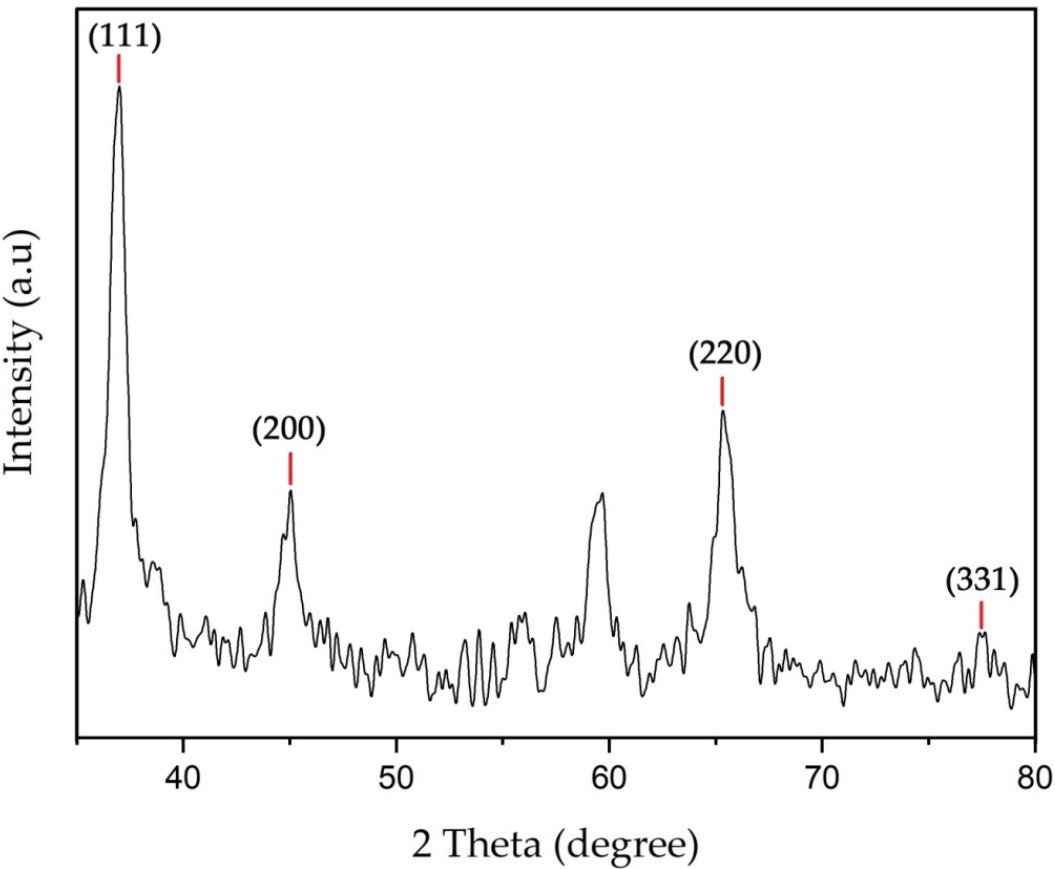

**Figure 12.** XRD pattern of the CS-coated GNS.

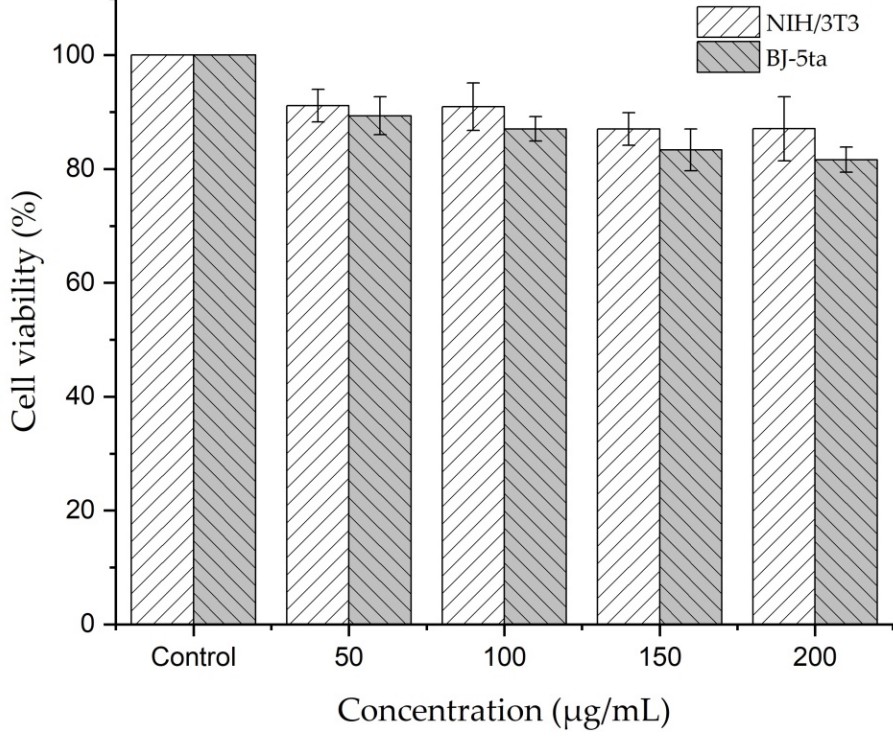

**Figure 13.** Cell viability of NIH/3T3 and BJ-5ta treated with GNS.

## 4. Conclusions

This study presents a rapid and green preparation of GNS using the seedless, free surfactant, and ultrasound-assisted method. The GNS particles obtained had long and sharp multi-branches with an average core of 67.85 ± 6.79 nm and a branched length of 76.11 ± 14.23 nm, through an adjusted conditional reaction, such as pH, mass concentration of CS, HQ concentration, as well as amplitude and time sonication. The influences of the conditions above and properties of the prepared GNS were characterized using UV–Vis absorption, FTIR, TEM, XRD, to determine the standard procedure for synthesizing GNS. Furthermore, cytotoxicity of the CS-coated GNS were investigated by the MTT assay on two cell lines, including NIH/3T3 and BJ-5ta. The results indicated that CS-coated GNS was a biocompatible agent, due to its high cell viability. Multi-branched gold nanoparticles are prospective materials not only for biomedical applications but also in cosmetics.

**Author Contributions:** Investigation, G.D.N., K.T.L.T. and T.M.H.; Investigation, writing—original draft preparation, P.T.H.; review and editing, V.Q.L.; writing—review and editing, project administration and funding acquisition, T.V.K.N. All authors have read and agreed to the published version of the manuscript.

**Funding:** This research was funded by Saigon Hi-tech Park under grant number 01/2020/HĐNVTX-KCNC-TTRD and the Project of Department of Science and Technology of Ho Chi Minh (118/2019/HĐ-QPTKHCN).

**Institutional Review Board Statement:** Not applicable.

**Informed Consent Statement:** Not applicable.

**Data Availability Statement:** All data is contained within the article.

**Conflicts of Interest:** The authors declare no conflict of interest.

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
