# Peer review of "Rapid and Green Preparation of Multi-Branched Gold Nanoparticles Using Surfactant-Free, Combined Ultrasound-Assisted Method"

_processes, doi:10.3390/pr9010112_

Round 1
Reviewer 1 Report
The authors described in details changes in the intensity of the surface plasmon resonance band and the position of its “maximum” for the star-like gold nanostructures (AuNSs) versus experimental conditions of their synthesis. However, there were no correlations found between these measures and the size of the AuNSs.
- The authors should give some relevant references to the literature, which explain the relation between the intensity and the position of the SPR band for such multi-branched gold nanostructures.
- How was it possible to determine the maxima of the SPR bands? The UV-Vis spectra were finished at 900 nm.
- It is difficult to see the cores in these structures as it is shown in TEM photographs. How many structures were taken for the determination of the size of these cores and branches? In reality, one figure with a high resolution TEM would be enough to clearly show the grain boundary.
- Tables 1-10. How many times were the UV-Vis of these AuNSs acquired?
- Were the AuNSs separated from the reaction mixtures prior to their analysis by UV-Vis and TEM?
- It seems that the synthesized AuNSs are not toxic to normal cell lines. Are they toxic to cancer cells? What could be their application? The authors have shown none of mentioned by them “biomedical applications”.
Author Response
Dear Professor - Reviewer
On behalf of members, I am so thankful for your review to help our manuscripts better. We rewrote and fixed the manuscripts as your comments.
Best regards.

Reviewer 2 Report
This paper reports a method for the preparation of multi-branched gold nanoparticles using hydroquinone as a reducing agent and the particles stabilized in chitosan. The cytotoxicity of the prepared and characterized particles were tested against normal rat 25 fibroblast (NIH/3T3) and normal human fibroblast (BJ-5ta) using the MTT assay. The aim of the authors in using the approach adopted for the preparation of the multi-branched gold nanoparticles was to produce material that could be readily used in biological applications. In its present form, this paper would require extensive editing to improve the language. In addition, there are major deficiencies for the attention of the authors and these include the following:
Main Comments
The introduction should be re-written to expand on the theory underpin the behavior of the multi-branched gold nanoparticles. See for example: dx.doi.org/10.1021/la2048097 | Langmuir 2012, 28, 8979−8984.
Provide an outline of methods that have been used to produce multi-branched gold nanoparticles and indicate how your method is different.
The UV/Vis spectra measured in this study do not clearly indicate the attachment of spikey protrusions from the spheroidal core. A TEM image of one of the particles would clearly show the detailed structure.
It is a good idea to dilute the samples before taking the UV/Vis spectra.
Explain the LSPR of the particles based on the structures established in the TEM measurements.
The Legends to the figures have to provide enough information so that it is clear what message you want to get across to the reader without reference to the main text.
Author Response
Dear Professor - Reviewer
On behalf of author's members, I am so thankful for your review to help our manuscripts better. We has rewrote the manuscripts and fixed mistake as your comments.
Best regards.

Reviewer 3 Report
The gold nanoparticles (Au NPs) with different shape are very important especially in the electronic and medical application. Therefore, the experiments about to change shape and size of Au NPs should be done. In the paper entitled „Rapid and Green Preparation of Multi-branched Gold Nanoparticles using Surfactant-free, Combined Ultrasound assisted Method” authors showed gold nanostars synthesized by green synthesis method. I recommend major revision of this manuscript after answers to comments:
- In materials and methods section: What was the speed of UV-Vis measurements? The FTIR spectra were collected using with technigues? Authors used powder or water solution of nanoparticles? If water solution, what was the beckgraound? Were the spectra smoothed, normalized? Authors used baseline correction of FTIR spectra? All these information should be described.
- UV-VIs should be done to 1100 nm, because in figures 1, 2 and 9 the maximum absorbance ofsome samples were not visible.
- In the paper International journal of molecular sciences 20 (20), 5011, 2019 Authors obtained nanostar gold nanoparticles with different size of arms. Does any parameter which control size of arms or core of gold nanostar in your synthesis method?
Author Response

(The authors gave the same response as above.)

Round 2
Reviewer 1 Report
The authors have improved their work, however, left two last questions from the previous review unanswered. Therefore, the readers still do not known two important things:
- Were the resultant AuNSs separated from the reaction mixtures prior to their analysis by UV-Vis and TEM as well as in the cell viability study.
- Why were such star-like AuNPs synthesized? In this place, the authors should mention the potential biomedical applications of the AuNSs.
Author Response
Response to Reviewers
December 18th, 2020
Dear Professor – Reviewer of Manuscripts Processes 1008808
On behalf of authors, I am so thankful for your precious time in reviewing our manuscripts and give us valuable comments to help our paper better.
Below we provide the point-by-point responses. Besides, all modifications in the manuscript have been underlined and highlighted in red.
Sincerely,
Thanh V. K. Ngo, PhD
Research Laboratories of Saigon Hi-tech Park.

Reviewer 2 Report
See attached file.

Author Response
December 18th, 2020
Dear Professor – Reviewer of Manuscripts Processes 1008808
On behalf of the authors, I appreciate you for your precious time in reviewing our manuscript. Your comments are so valuable that led to improve our current manuscript. We hope the manuscript after careful revisions meet your high standards. The authors welcome further constructive comments if any
Below we provide the point-by-point responses. Besides, all modifications in the manuscript have been underlined and highlighted in red.
Sincerely,
Thanh V. K. Ngo, PhD
Research Laboratories of Saigon Hi-tech Park.

Reviewer 3 Report
I accept revision version of manuscript.
Author Response
Response to Reviewers
December 18th, 2020
Dear Professor – Reviewer of Manuscripts Processes 1008808
On behalf of the authors, I am so thankful for your precious time in reviewing our manuscripts and give us valuable comments to help our paper better.
Below we provide the point-by-point responses. Besides, all modifications in the manuscript have been underlined and highlighted in red.
Sincerely,
Thanh V. K. Ngo, PhD
Research Laboratories of Saigon Hi-tech Park.

Round 3
Reviewer 2 Report
The present version of this paper has incorporated the recommendations I made in my second review but have not addressed the following comment: “As can be seen from the electron micrographs of the multi-branched nanoparticles and examination of their spectra, the dominate feature can assigned to the interaction of the spherical core of the particles with each other and with limited contribution from the branches. In other words, there is more contribution to the spectrum due to aggregation than from the surface plasmon resonance from the branches. The reported data should be re-interpreted in view of these observations.”
This comment is important for the interpretation of the data, and will help readers to appreciate what is shown in the SPR spectra.
In addition, the language and presentation need attention. I have re-written the abstract to improve clarity but the main body has to be carefully edited so that the results and discussion are clearly presented.
Abstract
The conventional seed-mediated preparation of multi-branched gold nanoparticles (GNS) use either cetyltrimethylammonium bromide or sodium dodecyl sulfate. However, both surfactants are toxic to cells so they have to be removed before the GNS can be used in biomedical applications. In this work, a green and facile method for the preparation of multi-branched gold nanoparticles using hydroquinone as a reducing agent in the presence of chitosan as stabilizer with ultrasound (US) irradiation used to improve the multi-branched shape and stability is described. The influence of pH, mass concentration of chitosan, hydroquinone concentration as well as sonication conditions such as amplitude and time of US on the growth of the multi-branched gold nanoparticles were investigated. The spectra of the GNS showed a broad band from 500 to over 1100 nm an indication of the effects of both aggregation and contribution of the multi-branches to the surface plasmon resonance (SPR) signal. Transmission electron microscopy (TEM) measurements of the GNS under optimum conditions(list the conditions here) showed average core diameter of 64.85 ± 6.79 nm and 76.11 ± 14.23 nm of the branches of multi-branched particles. Fourier Transfer Infrared Spectroscopy (FTIR) was employed to characterize the interaction between colloidal gold nanoparticles and chitosan, and the results showed the presence of the latter on the surface of the GNS. The cytotoxicity of chitosan capped GNS was tested on normal rat fibroblast (NIH/3T3) and normal human fibroblast (BJ-5ta) by MTT assay concentrations from 50-125 µg/mL with no adverse effects on cell viability.
Author Response
Dear Professor – Reviewer of Manuscripts Processes 1008808
On behalf of the authors, I appreciate you for your precious time in reviewing our manuscript. We had re-written the abstract according your comment as well as re-edited the main body of manuscript to re-interpret based on the effect of interaction of spherical core and plasmon resonance of branches on UV-Vis spectra as you commented. Your comments are so valuable that led to improve our current manuscript. We hope the manuscript after careful revisions meet your high standards. The authors welcome further constructive comments if any.
Below we provide the point-by-point responses. Besides, all modifications in the manuscript have been underlined and highlighted in red.
Sincerely,
Thanh V. K. Ngo, PhD
Research Laboratories of Saigon Hi-tech Park.
